# Evaluation of Wastewater Discharge Reduction Scenarios in the Buenaventura Bay

**Francisco-Fernando García-Rentería [1],\*, Gustavo Ariel Chang Nieto [2] and Gustavo Hernández Cortez [2]**

1   Gestión Evaluación y Monitoreo Ambiental—GEMA, Santa Marta 470004, Colombia
2   Department of Civil Engineering, Magdalena University, Santa Marta 470004, Colombia
*   Correspondence: fcofdogarcia@gmail.com

**Abstract:** Buenaventura Bay is facing severe pollution due to the direct discharge of untreated wastewater from 695 outlets along the coast, which serves 500,000 people. To address this issue, a study was conducted using the RMA11 water quality model, which was previously calibrated and validated, coupled with the RMA10 3D hydrodynamic model to assess sanitation scenarios in the bay. Five effluent reduction scenarios were proposed and compared based on fecal coliform concentration as an indicator, with evaluation also based on areas where fecal coliform concentrations exceeded the standard for primary contact. The model results revealed poor water quality in the bay, indicating that immediate action was necessary to prevent further deterioration. The proposed staged reduction in discharges would initially have more severe effects than the current situation, but this would improve when the treatment plant became operational. However, even with the plant in operation, the complete sanitation of the bay cannot be achieved, and further measures are required. This study emphasizes the urgent need for effective and sustainable measures to improve water quality in Buenaventura Bay and demonstrates the usefulness of the modeling approach in identifying effective sanitation scenarios to achieve this goal. The results highlight the need for a comprehensive management strategy to tackle pollution in the bay and provide insights for other regions facing similar challenges.

**Keywords:** hydrodynamic model; water quality model; Colombian Pacific coast; marine pollution; fecal coliforms

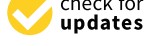



## 1. Introduction

Buenaventura Bay (Figure 1) has the second most important container port in Colombia, which operates and handles a significant volume of cargo. This logistics hub is responsible for nearly 45% of Colombia's international maritime cargo and plays a critical role in the country's economy. In 2020, Colombia exported over 99 million tons of cargo and received approximately 35 million tons and 61,000 ships on both the Atlantic and Pacific coasts, resulting in a total of 2 million containers over the year. Buenaventura is now the second–largest regional port company in the country and a key player in Colombia's maritime export routes [1].

Despite being home to Colombia's most important port on the Pacific, Buenaventura is currently a municipality with the highest levels of monetary and multidimensional poverty. Its residents face poor socioeconomic conditions and underdevelopment. According to the results of the continuous household survey (ECH) conducted by the National Administrative Department of Statistics (DANE), 62.7% of the population lives in poverty, and 20.9% are considered indigent. The coverage of basic services, particularly potable water, is 73.2%, while sanitation is at 61.0% [2]. With a population of 500,000, it is the most important settlement on the Colombian Pacific coast, with a diverse ethnic composition that includes indigenous and mestizo Afro-descendants. Furthermore, 30% of the population resides in stilt houses and palafitic dwellings, which are located in areas that are classified as public property under Colombian law [3,4].

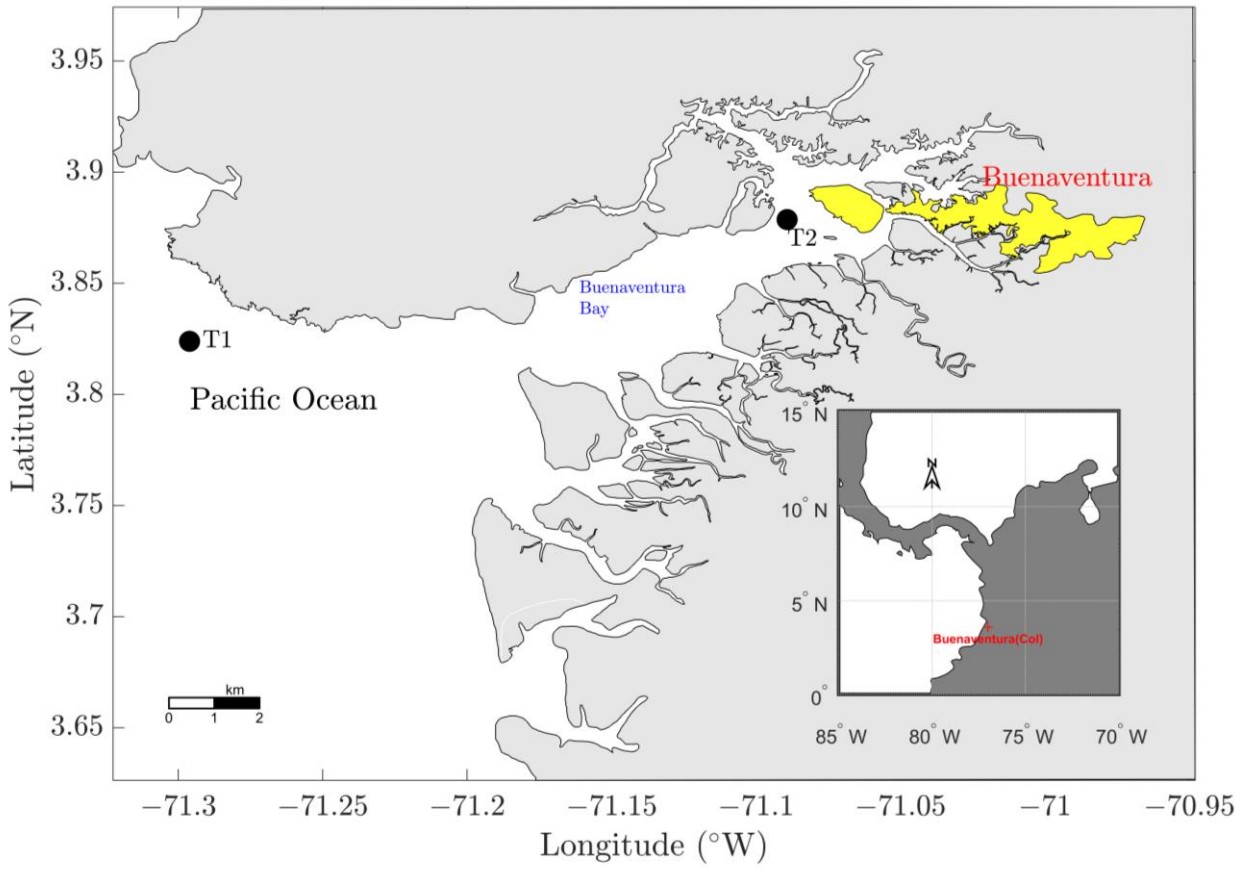

**Figure 1.** Buenaventura Bay localization.

The Bay of Buenaventura is considered a true estuary due to its semi-enclosed body of water with a connection to the sea and contributions of fresh water from the Dagua and Anchicaya rivers. Chemical substances from economic and domestic activities in the basins and bay have deteriorated the quality of the water and ecosystems. Evidence of contamination can be observed through indicator organisms in planktonic and benthic communities above the water column and mangroves [5]. This body of water has an average temperature of 26.4 °C, ranging from 23.3 °C to 33.5 °C. The average salinity varies slightly between climatic seasons; it is slightly higher in low rainfall seasons (13.0 ± 8.3 Practical Salinity Units—PSU) compared to high rainfall seasons (11.0 ± 5.8 PSU), which confirms its estuarine characteristics. The historical average of dissolved oxygen (DO) was 6.08 ± 1.09 mgO$_2$/L, and biochemical oxygen demand (BOD) averaged 1.58 ± 1.75 mgO$_2$/L. Domestic wastewater contamination is evident in the bay, with thermotolerant coliforms (>200 MPN/100 mL) and total coliforms (>5000 MPN/100 mL) [6].

The discharge of untreated wastewater poses a serious problem for both the environment and public health, affecting not only rivers and streams but also coastal areas, which reduces the availability of water resources and restricts their use. In Colombia, it is estimated that 95% of domestic and 85% of industrial wastewater was discharged without proper treatment, with 95% of agricultural wastewater also being released without treatment [7]. In Buenaventura, the sewage system has 695 outfalls that are discharged directly into the natural environment without any prior treatment, resulting in an estimated daily domestic wastewater production of 61,164 m$^3$ being discharged into Buenaventura Bay [8]. The estimated pollutant load of the untreated wastewater that is discharged into the bay is 2925 t/year of organic matter in the form of BOD, 702 t/year of dissolved inorganic nitrogen, 47 t/year of dissolved inorganic phosphorus, 2925 t/year of total suspended solids (TSS), and $1.2 \times 10^{19}$ t/year of coliform bacteria. The main tributaries provide a flow

of 345 m$^3$/s to the bay, carrying 6734 t/year of dissolved inorganic nitrogen, 12,298 t/year of BOD, and 5.81 × 10$^{19}$ t/year of thermotolerant coliform bacteria [9].

To address this issue, a plan has been proposed to manage and sanitize wastewater discharge (PSMV) in Buenaventura, with the goal of reducing and eliminating the 695 untreated discharges. However, the implementation of the plan is expected to take 30 years to complete. To achieve this reduction in untreated discharges, new sewage infrastructure and a wastewater treatment plant need to be constructed. During the construction period, the number of discharges will decrease, and there will be a temporary concentration of the discharged flow in certain areas before the treatment plant becomes operational.

When evaluating the impact of water pollution control measures on water bodies, forecasting introduces a high level of uncertainty. To improve the accuracy and reduce uncertainty in these forecasts, water quality models are commonly used [10]. These models are designed to predict the movement and dispersion of contaminants in water bodies [11] and are utilized in various water resource applications, including environmental impact assessments, pollution management, and remediation [12]. The effectiveness of water quality models in assessing future scenarios for managing and reducing contaminant discharges in water bodies has also been demonstrated by numerous other researchers [13–15].

The objective of this article is to determine the impact of sanitation solutions in Bahía de Buenaventura and provide decisions that can support in terms of load reduction analysis. The sanitation measures have been projected in stages that allow for the reduction and unification of discharges until the construction of two treatment plants is complete. Hydrodynamic and water quality modeling were used to analyze the temporal and spatial evolution of these sanitation measures, providing decision-makers with tools to determine the necessary speed with which they should be implemented.

## 2. Materials and Methods

### 2.1. Study Area

Buena Ventura Bay is located on the Colombian Pacific coast, with geographical coordinates ranging from 77.26° to 77.35° W and from 3.71° to 3.92° N (Figure 1). The bay has an elongated and narrow configuration, stretching approximately 21 km in length and with a width that varies between 3 and 11 km. It has a single entrance, known as La Bocana, which is formed by a 1.6-km-wide strait. The bay has an outer section that directly connects to the open sea and receives the influence of tides and currents and an inner section with estuary characteristics, where fresh water from various tributaries is discharged. This results in an average flow of 345 m$^3$/s, primarily from the Dagua and Anchicayá rivers, as well as the Pichidó, San Joaquín, Aguadulce, Gamboa, San Antonio, and Aguacate estuaries.

### 2.2. Wastewater Characterization

To assess the quality of the wastewater discharged into Buenaventura Bay, a characterization campaign was conducted. Fecal coliforms (FC) were determined following the standard methods for the examination of water and wastewater [16]. The 695 wastewater discharge locations were georeferenced, and a sample of 100 (14.39%) of these was selected for characterization. Figure 2 shows the location of the wastewater discharges.

Based on the characterization data, the discharges were categorized into three groups based on a low, medium, and high flow, and an average flow was calculated for each category. An estimated total wastewater discharge flow of 694 L/s was determined as flowing into Buenaventura Bay. Table 1 displays the flows used for each category of wastewater discharge in the simulations.

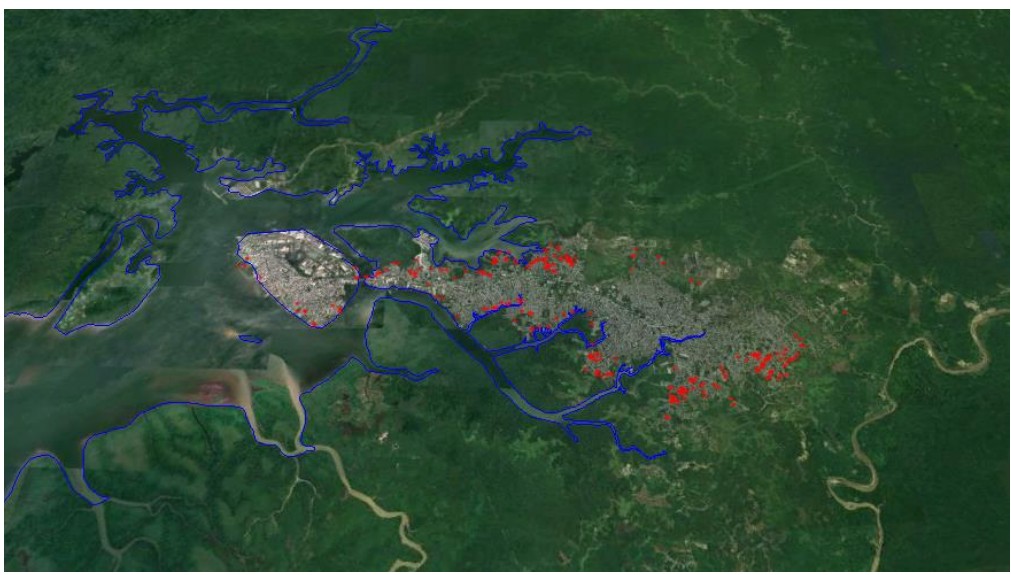

**Figure 2.** Location of wastewater discharges (red dots); the blue line represents the coastline, Google Earth image.

**Table 1.** Discharge flows that were considered in the simulations.

| Discharge Type | Mean Flow (L/S) | Flow Range (L/S) | Number of Discharge Points | Total Flow (L/s) |
|---|---|---|---|---|
| Low | 0.74 | 0.11–0.93 | 316 | 233.84 |
| Medium | 1.01 | 0.94–1.42 | 252 | 254.52 |
| High | 1.62 | 1.43–1.83 | 127 | 205.74 |

*2.3. Hydrodynamic and Water Quality Models Description*

A one-way coupling scheme was used to couple the hydrodynamic model (RMA10) and the water quality model (RMA11), as shown in Figure 3. The RMA10 model simulates the velocity, pressure, and sediment fields in three dimensions [17–20]. Equations (1)–(6) describe the system of equations that the model solves.

The equations of movement in the three components of the cartesian field are shown in Equations (1)–(3):

$$\rho\left(\frac{\partial u}{\partial t}+u\frac{\partial u}{\partial x}+v\frac{\partial u}{\partial y}+w\frac{\partial u}{\partial z}\right)-\frac{\partial}{\partial x}\left(\varepsilon_{xx}\frac{\partial u}{\partial x}\right)-\frac{\partial}{\partial y}\left(\varepsilon_{xy}\frac{\partial u}{\partial y}\right)-\frac{\partial}{\partial z}\left(\varepsilon_{xz}\frac{\partial u}{\partial z}\right)+\frac{\partial p}{\partial x}-\Gamma_x=0, \tag{1}$$

$$\rho\left(\frac{\partial v}{\partial t}+u\frac{\partial v}{\partial x}+v\frac{\partial v}{\partial y}+w\frac{\partial v}{\partial z}\right)-\frac{\partial}{\partial x}\left(\varepsilon_{yx}\frac{\partial v}{\partial x}\right)-\frac{\partial}{\partial y}\left(\varepsilon_{yy}\frac{\partial v}{\partial y}\right)-\frac{\partial}{\partial z}\left(\varepsilon_{yz}\frac{\partial v}{\partial z}\right)+\frac{\partial p}{\partial y}-\Gamma_y=0, \tag{2}$$

$$\rho\left(\frac{\partial w}{\partial t}+u\frac{\partial w}{\partial x}+v\frac{\partial w}{\partial y}+w\frac{\partial w}{\partial z}\right)-\frac{\partial}{\partial x}\left(\varepsilon_{zx}\frac{\partial w}{\partial x}\right)-\frac{\partial}{\partial y}\left(\varepsilon_{zy}\frac{\partial w}{\partial y}\right)-\frac{\partial}{\partial z}\left(\varepsilon_{zz}\frac{\partial w}{\partial z}\right)+\frac{\partial p}{\partial z}+\rho\cdot g-\Gamma_z=0 \tag{3}$$

The continuity equation is presented in Equation (4):

$$\frac{\partial u}{\partial x}+\frac{\partial v}{\partial y}+\frac{\partial w}{\partial z}=0 \tag{4}$$

Advection–diffusion processes are presented in Equation (5):

$$\frac{\partial s}{\partial t}+u\frac{\partial s}{\partial x}+v\frac{\partial s}{\partial y}+w\frac{\partial s}{\partial z}-\frac{\partial}{\partial x}\left(D_x\frac{\partial s}{\partial x}\right)-\frac{\partial}{\partial y}\left(D_Y\frac{\partial s}{\partial y}\right)-\frac{\partial}{\partial z}\left(D_z\frac{\partial s}{\partial z}\right)-\theta s=0, \tag{5}$$

Additionally, the equation of state is presented in Equation (6):

$$\rho=F(s), \tag{6}$$

where $x$, $y$, and $z$ are the coordinates of the Cartesian system; $u$, $v$, and $w$ are the velocities in the directions of the Cartesian system; $p$ is the water pressure, $t$ is time, $\varepsilon_{xx}$, $\varepsilon_{xy}$, $\varepsilon_{xz}$, $\varepsilon_{yx}$, $\varepsilon_{yy}$, $\varepsilon_{yz}$, $\varepsilon_{zx}$, $\varepsilon_{zy}$, $\varepsilon_{zz}$ are the eddy turbulence coefficients, $g$ is the acceleration due to gravity, $D_x$, $D_Y$, and $D_z$ are the eddy diffusion coefficients, $\rho$ is the density of the water, $\Gamma_x$, $\Gamma_y$, and $\Gamma_z$ are the external forces; $s$ is the salinity, and $\theta s$ is the salinity source/sink.

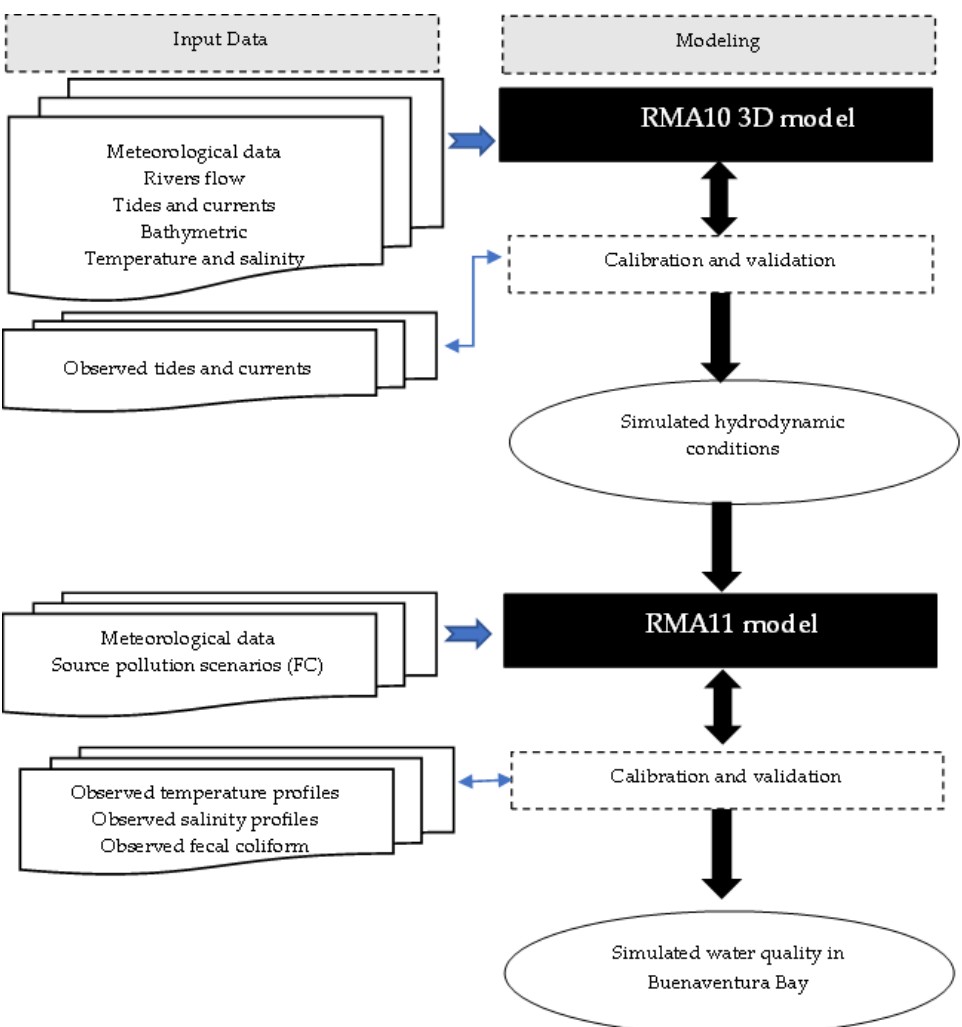

**Figure 3.** One-way coupling scheme for RMA10 and RMA11 models.

The RMA10 model is a widely applicable hydrodynamic modeling tool for various water bodies, including lakes, bays, estuaries, and rivers. Recent studies have demonstrated its effectiveness in flood management, sediment transport modeling, flow dynamics simulation, water quality assessment, and thermal environment simulations. Specifically, Xiong et al. [21] used the RMA10 3D model to simulate a complex flood event in an urban area of China, while Ma et al. [22] applied it to simulate sediment transport in the Yangtze River. Wang et al. [12] analyzed water level and flow velocity in a Chinese river with a complex terrain using the RMA10 model, and Zhang et al. [23] used the RMA10 3D model to simulate water quality in a river in China. Li et al. [24] showcased the RMA10 model's potential to effectively simulate thermal dynamics in a river and provide essential information for the management of river ecosystems and the protection of aquatic organisms.

The RMA11 model, which uses the finite element method, has been successfully applied to model water quality in estuaries, bays, lakes, and rivers [25–28]. It receives the velocity, temperature, and salinity fields from the RMA10 model and uses them to solve advection-diffusion constituent transport equations [25]. The water quality relationships implemented in RMA11 are derived from QUAL2E. For more information, readers are

referred to Brown and Barnwell [29]. In RMA11, coliform transport is modeled using three loss parameters: settling, decay in darkness, and light-sensitive decay. The equation for coliform growth (GC) is as follows:

$$GC = -(K_{C1} + K_{C2} + K_{C3}/d) \times C_C \tag{7}$$

where $C_C$ is the concentration of the coliform (MPN/100 mL), $K_{C1}$ is the coliform die-off rate in darkness-temperature adjusted (1/day); $K_{C2}$ is the coliform die-off rate due to light-temperature adjusted (1/day); $K_{C3}$ is the coliform settling rate-temperature adjusted (m/day) and $d$ is the water depth (m).

The temperature values computed in RMA10 were used to adjust the rate coefficients in the source/sink terms. These coefficients were input at 20 °C and then corrected to the actual temperature ($T$) using Equation (8):

$$X_t = X_{20}\theta^{(T-20)} \tag{8}$$

where $X_t$ is the value of the coefficient at the local computed temperature, $X_{20}$ is the value of the coefficient at 20 °C, and $\theta$ is a dimensionless constant coefficient usually set to 1.07 [30]. This correction was applied to $K_{C1}$, and $K_{C3}$, while $K_{C2}$, which is dependent on the light intensity, is given in Equation (9):

$$K_{C2} = 2,3026\frac{\left[L_i exp(-\lambda z_d)^{0.7}\right]}{L_c} \tag{9}$$

where $L_i$ is the light intensity expressed in MJ/m²–h, $L_c$ is the coliform light coefficient (h·[MJ/m²–h]$^{0.7}$), $\lambda$ is the light extinction coefficient (1/m), and $z_d$ is the depth below the water surface in 3D (m).

The RMA11 water quality model is a highly adaptable and effective tool for evaluating water quality, identifying sources of pollution, and assessing the efficacy of pollution control measures in various aquatic systems. A number of previous studies have demonstrated its effectiveness in diverse applications, including the assessment of water quality and eutrophication [31], the impact of treated sewage discharge on coastal water quality [32], and the analysis of spatial and temporal variations in water quality [33]. The RMA11 model has also been successfully used to predict water quality and ecological responses in shallow lakes [34], simulate water quality in typical urban rivers [35–39], and evaluate the effectiveness of water environment protection measures [40]. Overall, these studies highlight the RMA11 model's versatility and effectiveness in evaluating water quality and pollution control measures in a wide range of aquatic systems.

*2.4. Model Set up for Buenaventura Bay*

The simulation domain was chosen to cover an area of 430 km², including both the inner and outer bay. The finite element mesh closely followed the bay's irregular coastline and was created using the Mesh2D software [41,42]. The resulting mesh comprised 19,672 elements, with lengths ranging from 15 to 500 m and 43,604 nodes. The mesh's triangular elements were equilateral, reducing the computational efforts. Figure 4 depicts the mesh generated by Mesh2D.

The bathymetry data used in the study were obtained from the CIOH nautical charts of the study area and digitized and interpolated at the finite element mesh nodes. The interpolated bathymetry is shown in Figure 5. The RMA10 model used sigma coordinates and modified the system of equations to account for changes in the water surface elevation due to tides. Details of the modification can be found in the references by Marthanty et al. [17] and Fosati et al. [18]. The finite element mesh was discretized into five layers in the vertical direction.

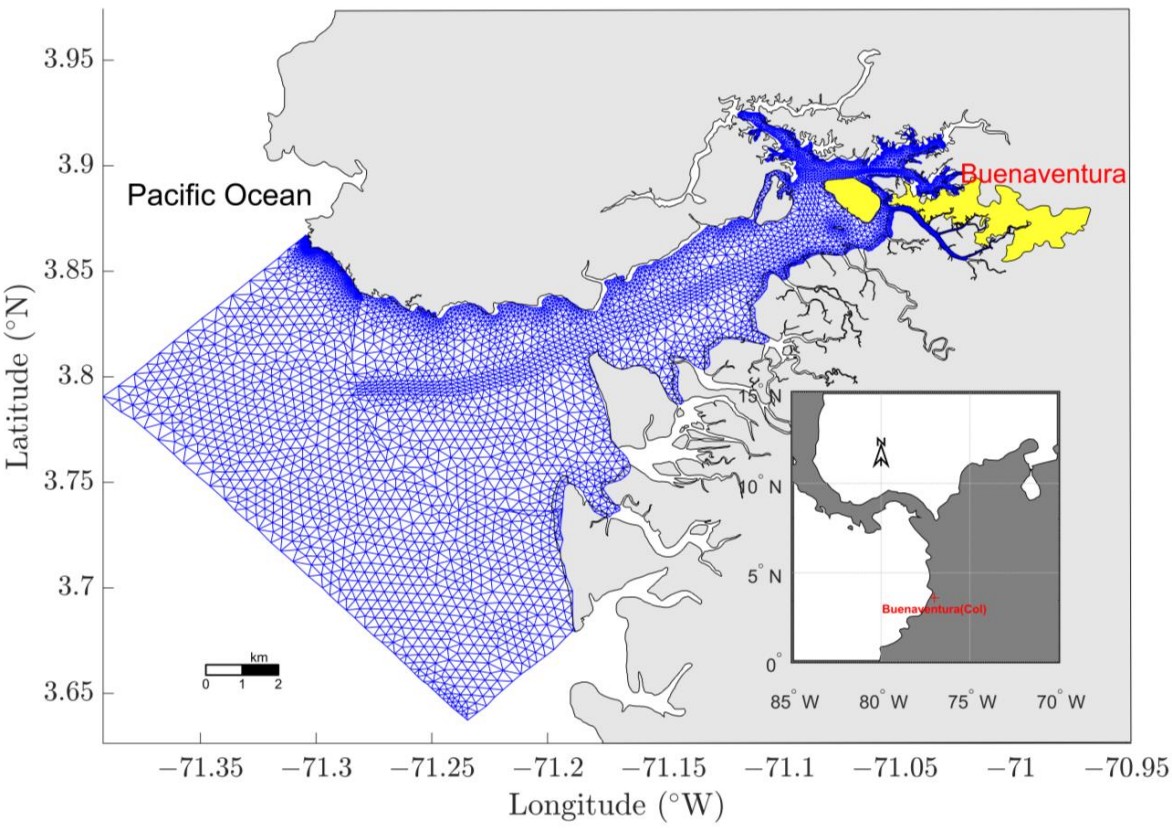

**Figure 4.** Buenaventura Bay finite element mesh (blue), urban area in yellow.

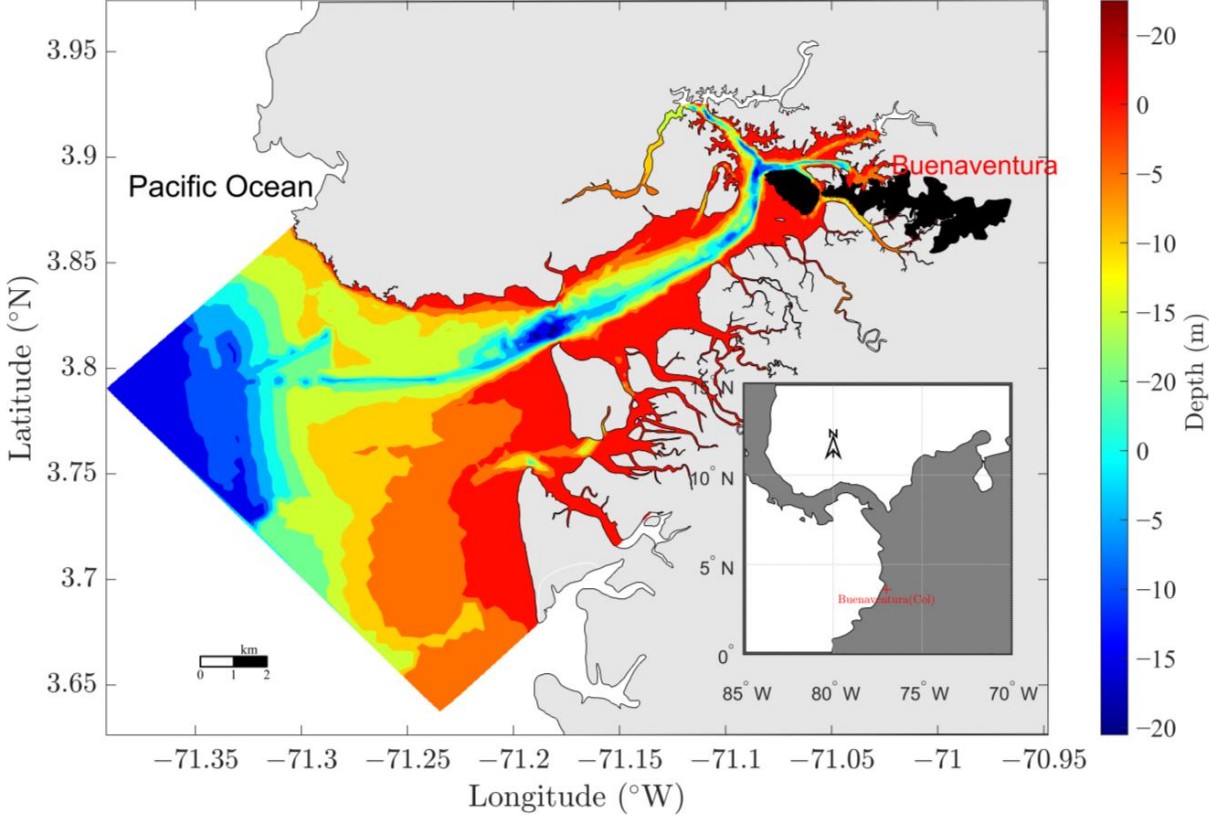

**Figure 5.** Simulation domain bathymetry built-up area with black.

The open boundaries at the southeast, southwest, and northwest of the outer bay were forced with the current, water level, salinity, and temperature data from the HYCOM global model [43]. Wind direction and speed, solar radiation, humidity, and air temperature data were taken from a meteorological station and considered uniform over the simulation domain. The simulation model was run for one year (2021) to determine the effect of wastewater discharges on Buenaventura Bay. Freshwater from the Dagua (66.10 m$^3$/s), Achicayá (98.90 m$^3$/s), Humane (30 m$^3$/s), Gamboa (30 m$^3$/s), Aguacate (10 m$^3$/s), San Antonio (20 m$^3$/s), Hondo (10 m$^3$/s), and Agua Dulce (80 m$^3$/s) rivers and estuaries was considered in the model. The mean flows and temperatures of each river and estuary were accounted for. Average FC concentrations of 28,960 and 4724 MPN/100 mL were included in the Dagua and Achicayá rivers, respectively, as reported by Vivas et al. [9].

*2.5. Calibration and Validation of Models*

The models were calibrated by comparing the simulation results with measurements of currents, tides, and fecal coliform (FC) concentrations (MPN/100 mL). The calibration period was from February to March, while the validation period was from October to November and was chosen to capture seasonal variations in rainfall. The region experiences a high rainfall of over 6500 mm/year and has two seasons with distinguishable rainy periods: a low rainy season from January to June and a high rainy season from July to December [44]. The monitoring station that is used for current and tidal measurements is indicated as T1 in Figure 1. AANDERAA RCM9 LW (0 to 300 cm/s) equipment was deployed at a depth of 6 m for current measurements, while tidal measurements were made with AANDERAA WLR7 equipment (0–700 kPa), and both devices were programmed to collect data every 15 min.

Weekly measurements of FC concentrations were taken at the measurement station marked as T2 in Figure 1. During the calibration period from February to March for the RMA11 water quality model, measurement campaigns for validation were conducted from October to November at the same location. FC measurements were taken at a depth of 1 m in the seawater column.

To assess the reliability of the models during calibration and validation, the root mean square error (RMSE) (Equation (10)) and *Skill* (Equation (11)) estimators were used:

$$\text{RMSE} = \left\{ \frac{1}{N} \sum_{i=1}^{N} [\beta_m - \beta_d]^2 \right\}^{1/2} \tag{10}$$

$$Skill = 1 - \frac{\sum[\beta_m - \beta_d]^2}{\sum(|\beta_m - \overline{\beta_d}| - |\beta_d - \overline{\beta_d}|)} \tag{11}$$

where $\beta_m$ is the measured parameter, $\beta_d$ is the model result parameter, N is the number of samples in the time series, and $\overline{\beta_d}$ is the mean value of the observation. *Skill* is a statistical metric that is used to evaluate the accuracy and performance of models. This measure assesses the ability of a model to make accurate predictions by comparing the statistics of the observed data with those generated by the model. A *Skill* value that is close to one indicates high accuracy, while a value of zero indicates no predictive ability, and a negative value implies a worse-than-random prediction [45]. A high *Skill* value suggests a reliable model that can be used to make accurate predictions. Conversely, a low *Skill* value indicates that the model requires improvement to make it more useful for water management and protection.

*2.6. Wastewater Discharge Reduction Scenarios*

The Buenaventura PSMV proposed the construction of sewerage infrastructure to eliminate untreated wastewater discharges into Buenaventura Bay. Given the construction costs, it must be developed in stages that include primary and secondary collectors to

transport the wastewater, two treatment plants, and a submarine outfall for the final disposal of the treated wastewater.

Five simulation scenarios were proposed to evaluate the PSMV of Buenaventura Bay (Table 2). Scenarios from one to four represent a series of sequential stages that should be developed to effectively reduce the 695 untreated wastewater discharges to only two effluents from wastewater treatment plants. These scenarios are not mutually exclusive and can be implemented in combination. A fifth scenario was proposed to consider a future situation in which no sanitation solution is implemented. This scenario assumes that the discharges persist over time and are further aggravated by demographic and urban growth in the region. Conversely, scenario five is mutually exclusive and serves as a baseline for comparison with the other scenarios against a hypothetical 30-year period during which no solution is implemented. This approach enables the identification of the most effective strategy to mitigate the environmental impact of untreated wastewater discharges in the Bay of Buenaventura. The information presented in this study will be useful for guiding decision–making processes for environmental management in the region. The location of discharges in scenarios two and three–four are presented in Figure 6. The location of the discharges in scenarios one and five corresponds to the current location and is shown in Figure 2.

**Table 2.** Simulation scenarios proposed to evaluate the PSMV of Buenaventura Bay.

| Scenario Number | Number of Discharges Considered | Time of Implementation | Description |
|---|---|---|---|
| One | 695 | Current Situation | A total of 695 discharges of untreated wastewater. |
| Two | 6 | 10 years | The collection of all discharges in the sewage system to pass to six unique discharges that concentrate the flow of these. Six new concentrated discharges of untreated wastewater. |
| Three | 2 | 15 years | The collection of six discharges in the sewage system to pass to two unique discharges that concentrate the flow of these. Two new concentrated discharges of untreated wastewater. |
| Four | 2 | 30 years | Two wastewater discharges, removing 90% of the wastewater contaminants with the treatment plants in operation. Two effluents of treated wastewater. |
| Five | 695 | 30 years | A future scenario without a sanitation solution for the Bay of Buenaventura, considering an increase in the discharge flow. A total of 695 discharges of untreated wastewater. |

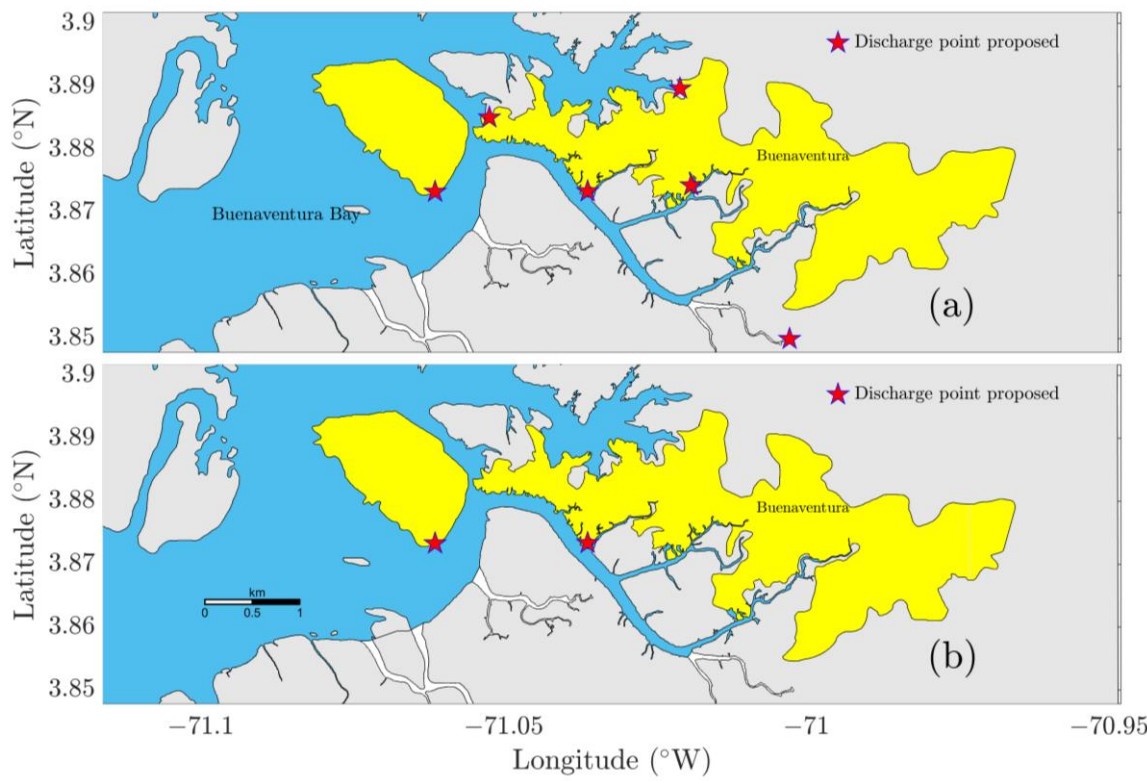

**Figure 6.** Discharge point in the scenarios proposed, (**a**) Scenario two; (**b**) Scenarios three and four.

## 3. Results

### 3.1. Characterization of Wastewater Discharges

The results of the FC concentrations of the wastewater discharge characterization into Buenaventura Bay are presented in Table 3. A total of 16 measurement campaigns were conducted to determine the FC concentration, with an average concentration of $8 \times 10^8$ MPN/100 mL observed.

**Table 3.** The concentration of FC in wastewater discharge.

| Parameter | Unit | Maximum | Mean | Minimum |
|---|---|---|---|---|
| Fecal Coliform | MPN/100 mL | $2 \times 10^{10}$ | $8 \times 10^8$ | $7 \times 10^3$ |

### 3.2. FC Concentration in the Water Column

Of the 16 measurement campaigns, eight were carried out during the calibration phase of the water quality model from February to March at a weekly frequency, and eight were conducted during the validation period from October to November. Mean values of $2.8 \times 10^3$ MPN/100 mL were found for the calibration stage and $1.8 \times 10^3$ MPN/100 mL for the validation stage. Table 4 displays the maximum, minimum, and average values of the sampling campaigns for each dataset.

**Table 4.** The maximum, minimum, and mean concentration of FC in seawater at Buenaventura Bay during the calibration and validation stages.

|  | Unit | Maximum | Mean | Minimum |
|---|---|---|---|---|
| Calibration | MPN/100 mL | $5 \times 10^3$ | $2.8 \times 10^3$ | $1.2 \times 10^3$ |
| Validation | MPN/100 mL | $2.4 \times 10^3$ | $1.8 \times 10^3$ | $4 \times 10^2$ |

The one-way ANOVA did not reveal significant differences ($p = 0.478$) between the two sample sets taken for the calibration and validation of the water quality model. Therefore, it can be presumed that, despite the difference in rainy seasons during which the samples were collected, the concentrations tended to be similar, given their origin from the discharge of wastewater into Buenaventura Bay.

### 3.3. Calibration Hydrodynamic Model

#### 3.3.1. Sea Level

The RSME estimator values were 0.602 and 0.620 during the calibration period (February–March) and validation period (October–November), respectively. The corresponding *Skill* values were 0.747 and 0.742 for the same periods. Figure 7 shows a comparison of the model results and field measurements for tides at station T1.

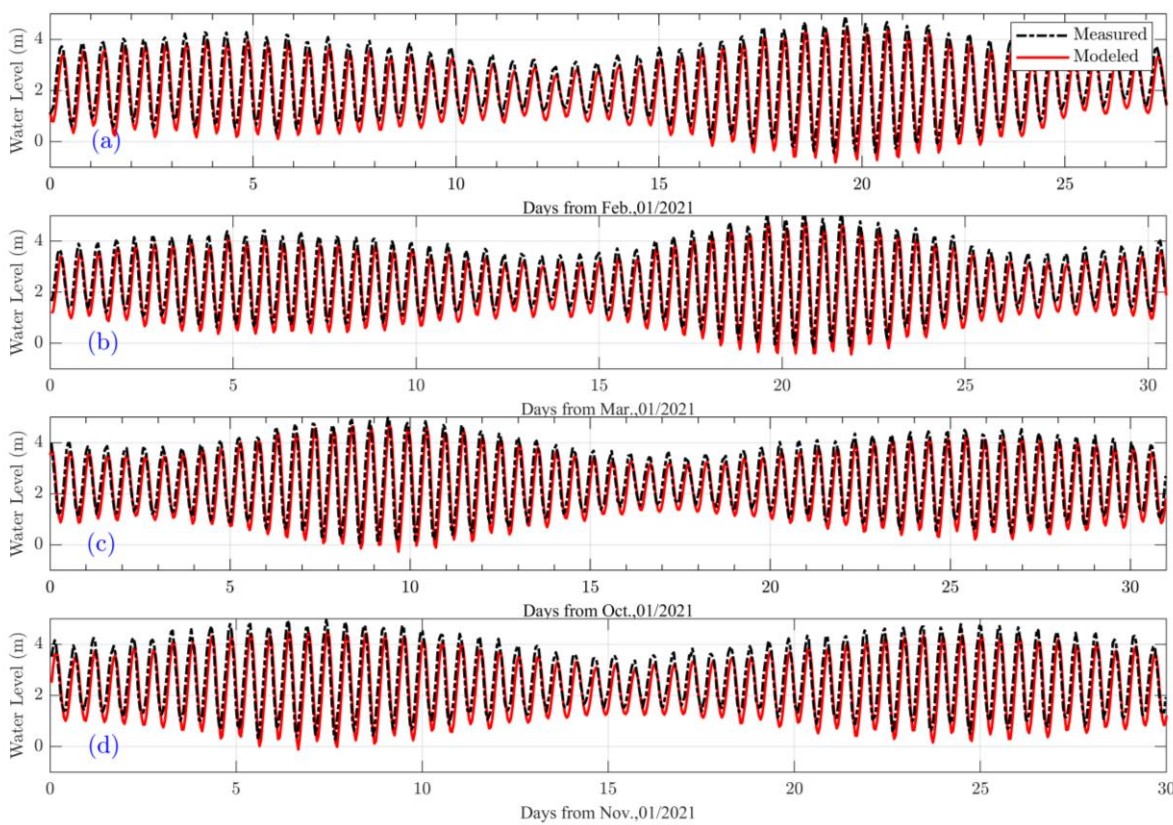

**Figure 7.** Sea water level comparison for simulations and measurements for (**a**) calibration in February, (**b**) calibration in March, (**c**) validation in October, and (**d**) validation in November.

#### 3.3.2. Currents

The performance estimators of the model were evaluated for model calibration, resulting in an RMSE of 0.007 m/s and a *Skill* score of 0.998 for current magnitude and an RMSE of 9.87 degrees and *Skill* score of 0.992 for the current direction, both indicating the excellent performance of the model in reproducing current data in the Bay of Buenaventura. The graphical comparison of the current roses for the measured field data and the model results confirms these estimators (see Figure 8).

Similar behavior was found in the evaluation of the model during the validation period. For the speed of the current, the RSME was 0.007 m/s, and the predictive ability was 0.998, while for the direction of the current, the values were 7.71 degrees and 0.995 for the RSME and *Skill*, respectively. The current roses resulting from the validation period (October–November) are shown in Figure 9; this corresponds to the comparison of the measured and simulated current data at station T1.

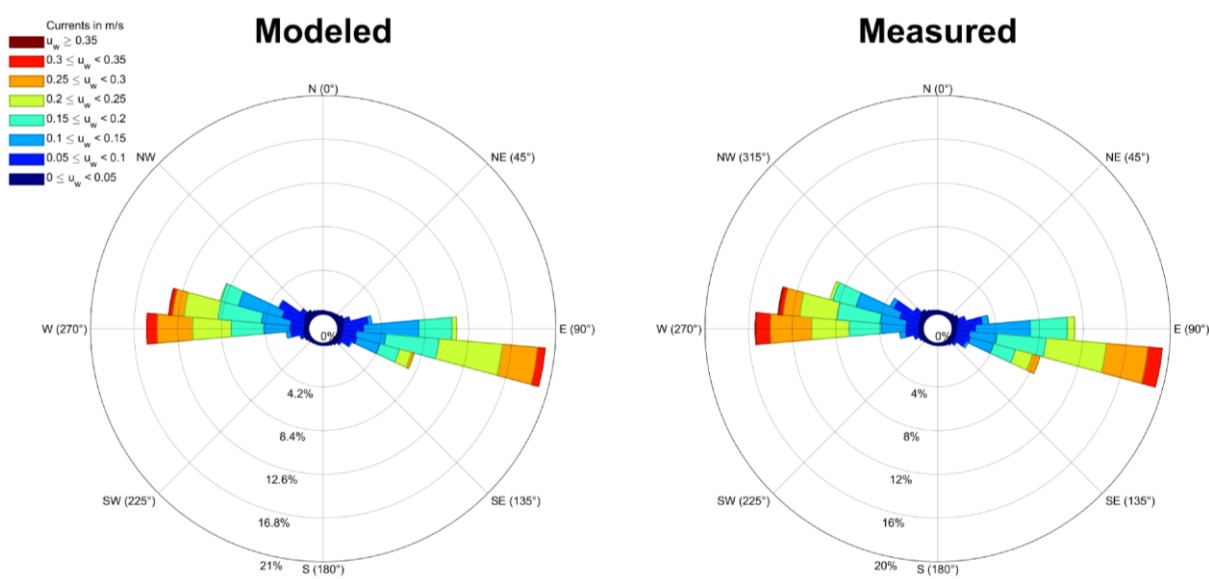

**Figure 8.** Comparison of the current roses for simulations and measurements of the calibration period.

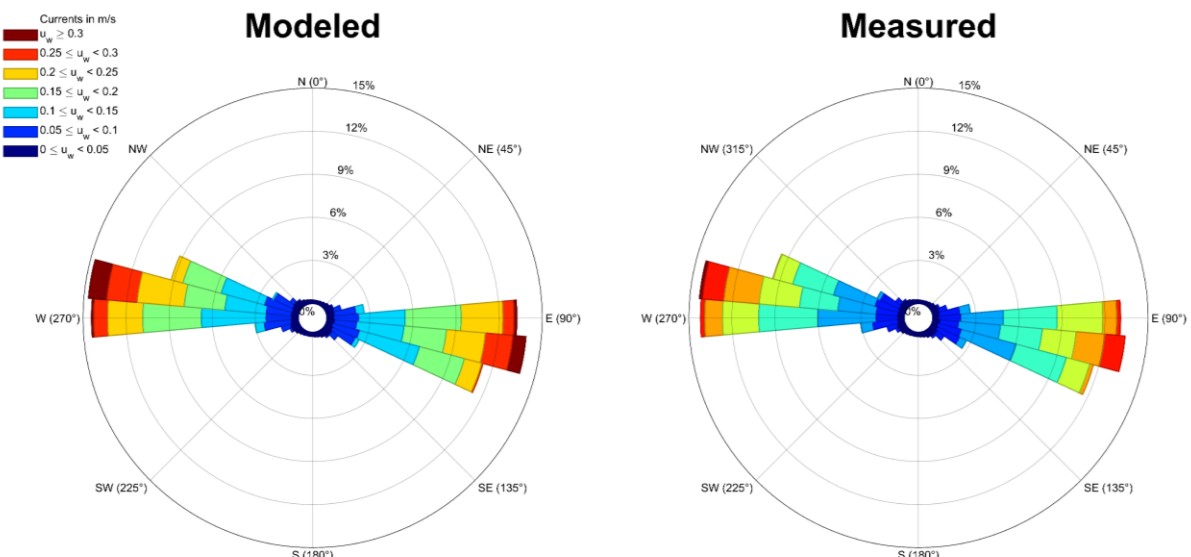

**Figure 9.** Comparison of the current roses for simulations and measurements for the validation period.

### 3.3.3. Water Quality Model

The water quality model exhibited a high predictive ability with values of 0.716 and 0.799 during the calibration and validation stages, respectively. Given the favorable results obtained, it was considered that the model could be capable of accurately predicting water quality data in Buenaventura Bay regarding FC. The RMSE values for the calibration and validation periods were $1 \times 10^3$ and $1.13 \times 10^3$, respectively. Figure 10 presents a graphical comparison between the field–measured data at station T2 and the model outcomes.

### 3.3.4. Discharge Reduction Scenario

Figure 11 displays the concentration of FC in the Bay of Buenaventura under scenario one, as estimated by the RMA11 model. This scenario investigates the effect of 695 untreated wastewater discharges, with the San Antonio estuary experiencing the highest concentrations of FC due to receiving most of the untreated wastewater discharges from Buenaventura. The highest recorded FC concentration in this scenario was $1.45 \times 10^7$ MPN/100 mL.

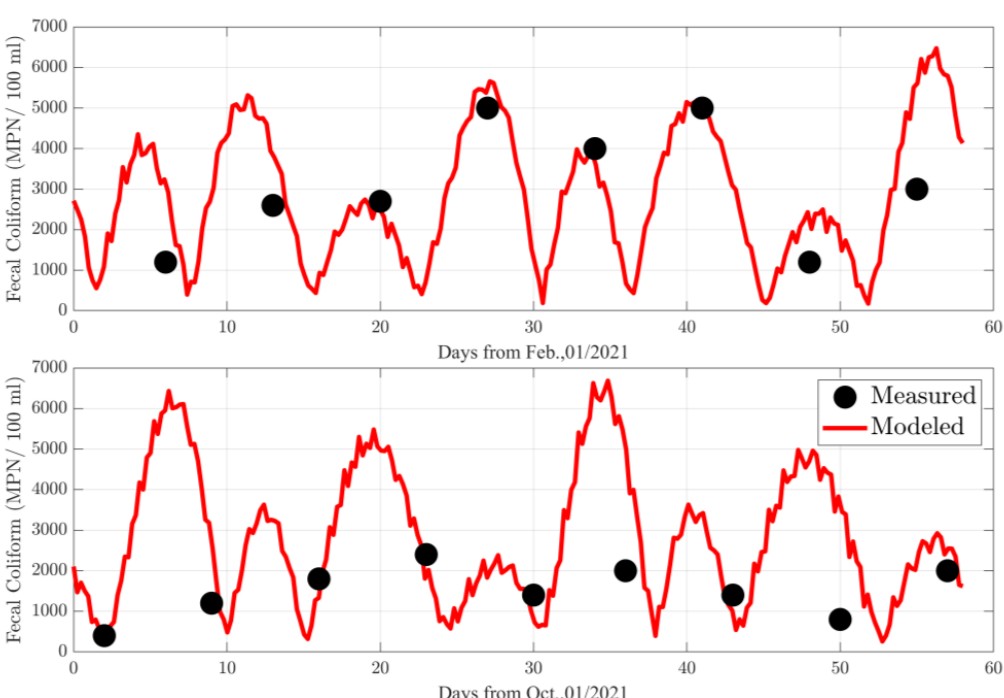

**Figure 10.** Comparison of fecal coliform concentration in the calibration and validation of the water quality model.

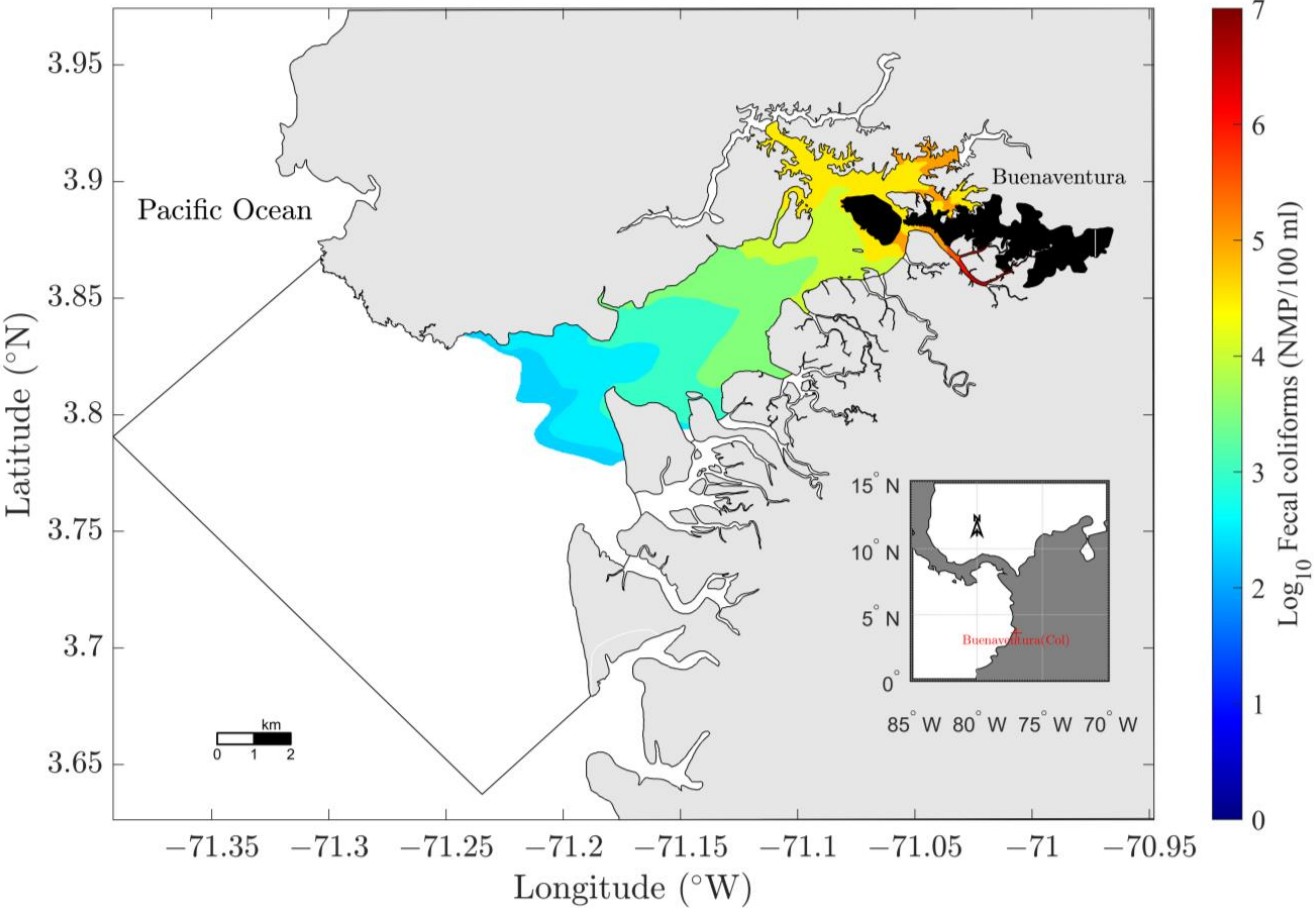

**Figure 11.** Spatial distribution of the maximum concentration of FC in the Bay of Buenaventura in the current scenario.

The concentration of FC in the water is a crucial indicator of the water's quality, especially for bodies of water that are used for primary contact activities, such as swimming and diving. To ensure public safety, the Colombian standard (Decree 1594 of 1984) set a maximum allowable limit of 200 MPN/100 mL in the water of primary contact activities. This limit was used as a reference for analyzing the impact of reducing discharge points in the Bay of Buenaventura. The water quality model employed a 15-min time step and was run for a year (2021), resulting in 35,040 data points per node. The analysis determined the frequency of affected areas with concentrations exceeding 200 MPN/100 mL for each node, as depicted in Figure 12. This analysis serves as the baseline scenario for comparing the outcomes of the proposed discharge reduction scenarios. The areas with a 100% frequency of values above 200 MPN/100 mL are highlighted in red in Figure 12.

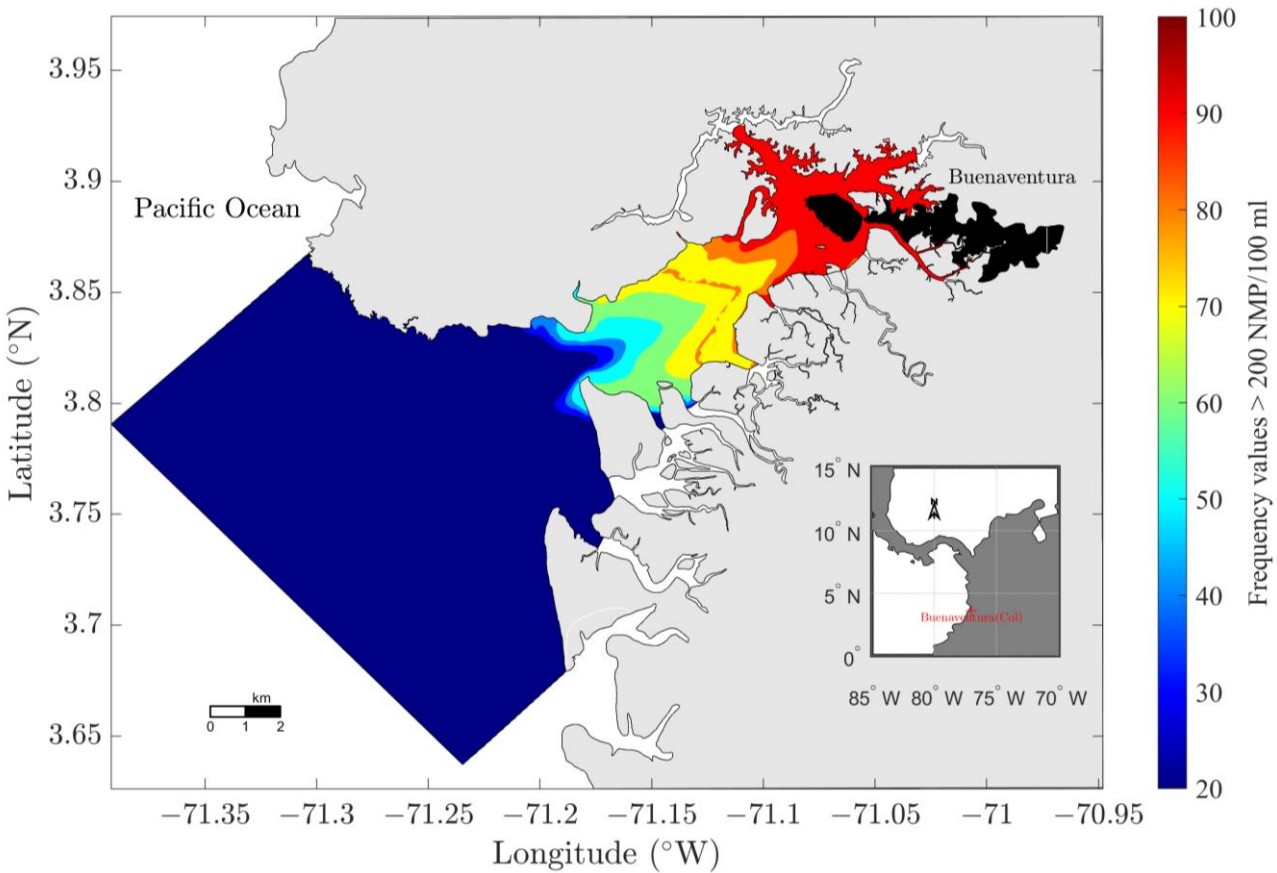

**Figure 12.** Frequency of areas affected by concentrations higher than 200 MPN/100 mL in the water column scenario one.

In order to evaluate the impact of urban wastewater discharges on the water quality in the Bay of Buenaventura, the water quality model was modified by setting the concentrations of FC in the tributaries that flow into the bay at zero. This modification enabled the impact of discharges generated solely by urban areas to be analyzed without interference from other sources. The adjustment parameters required for this modification were determined during the calibration and validation phase.

Under scenario one, the RMA11 model revealed a substantial impact of untreated wastewater discharges on the concentration of FC, with the highest levels observed in the estuary and near the discharge points. In most internal areas, FC concentrations consistently exceeded the maximum allowable limit, which occurred 100% of the time based on the model results (35,040 results with a time step of 15 min for a one-year run—2021). In the central part of the bay, high FC levels were observed between 50% and 70% of the time, while outside the bay, this impact was less severe, with concentrations exceeding the limit less than 20% of the time.

Scenario two examines the impact of reducing the number of wastewater discharge points from 695 to 6. The model results show an increase in FC concentrations in the bay, with a maximum concentration of $1.45 \times 10^7$ MPN/100 mL (Figure 13). In scenario three, the concentration of FC was found to be twice that of scenario two and 2.2 times that of scenario one when the untreated wastewater discharges were concentrated at two points. The maximum concentration of FC in scenario three was $3.21 \times 10^7$ MPN/100 mL (Figure 14).

To address the negative impact of untreated wastewater discharges, a wastewater treatment system was proposed for the two concentrated discharge points that were examined in scenario three. The implementation of the treatment system (scenario four) resulted in a significant reduction in the concentration of FC, with the maximum concentration reduced to $5.32 \times 10^5$ MPN/100 mL: the lowest among all the scenarios studied. The reduction in the FC concentration also led to a noticeable decrease in the affected area in Buenaventura Bay. Visual representations of the proposed scenarios and their outcomes are provided in Figure 15.

The last scenario (Figure 16) investigated the potential outcome if no measures were taken to address the issue of untreated wastewater in Buenaventura. The assumptions for this scenario take into account local demographic dynamics, including a population growth rate of 3% and the persistence of current discharges. Based on these assumptions, the projections indicate that the population of Buenaventura doubled over 25 years, resulting in the generation of 1.45 m$^3$/s of untreated wastewater.

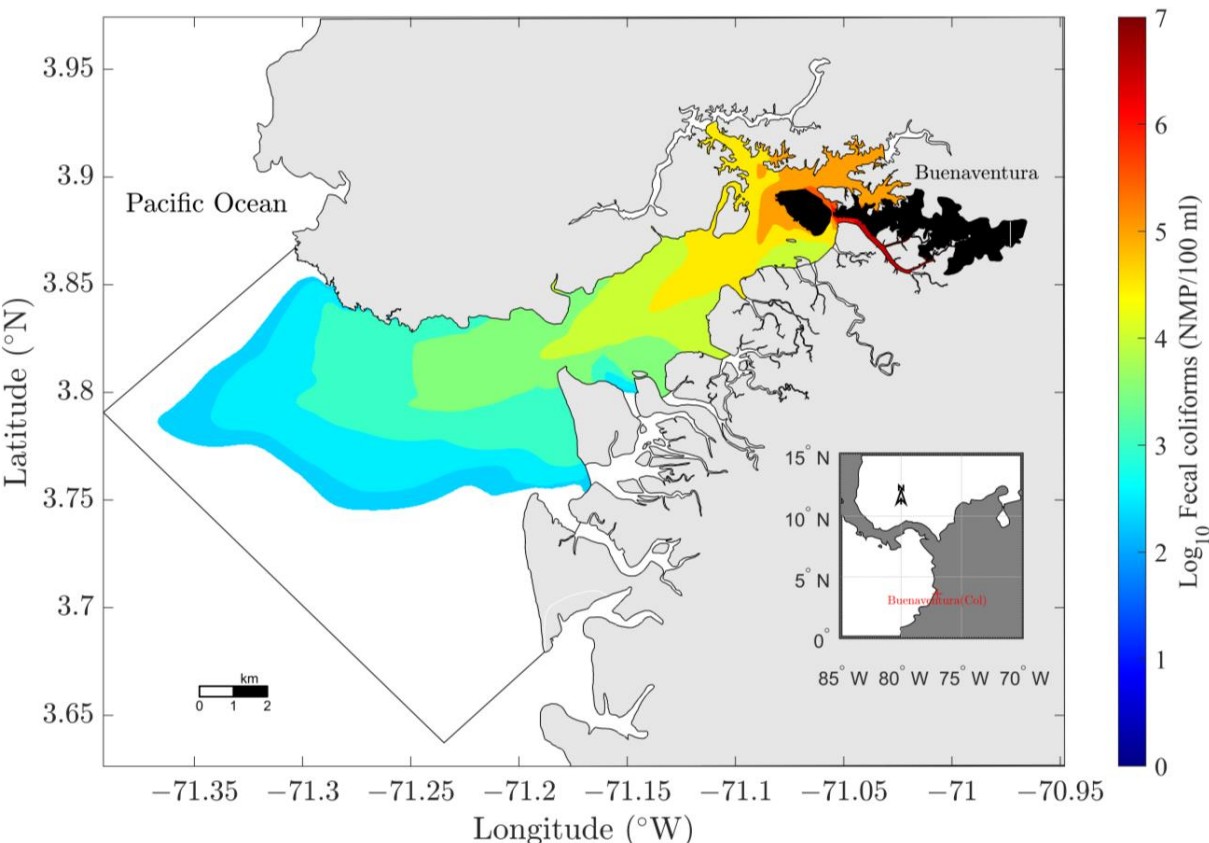

**Figure 13.** Spatial distribution of the maximum fecal coliform concentration in Buenaventura Bay in scenario two.

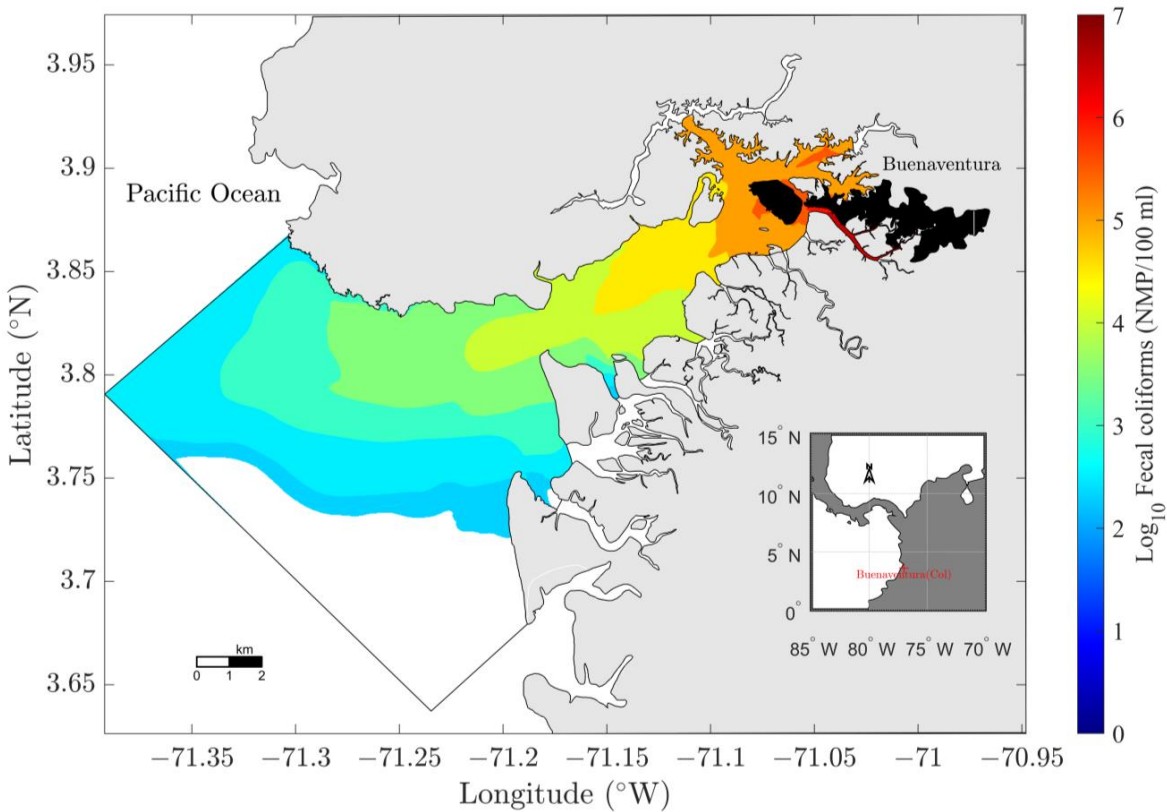

**Figure 14.** Spatial distribution of the maximum fecal coliform concentration in Buenaventura Bay in scenario three.

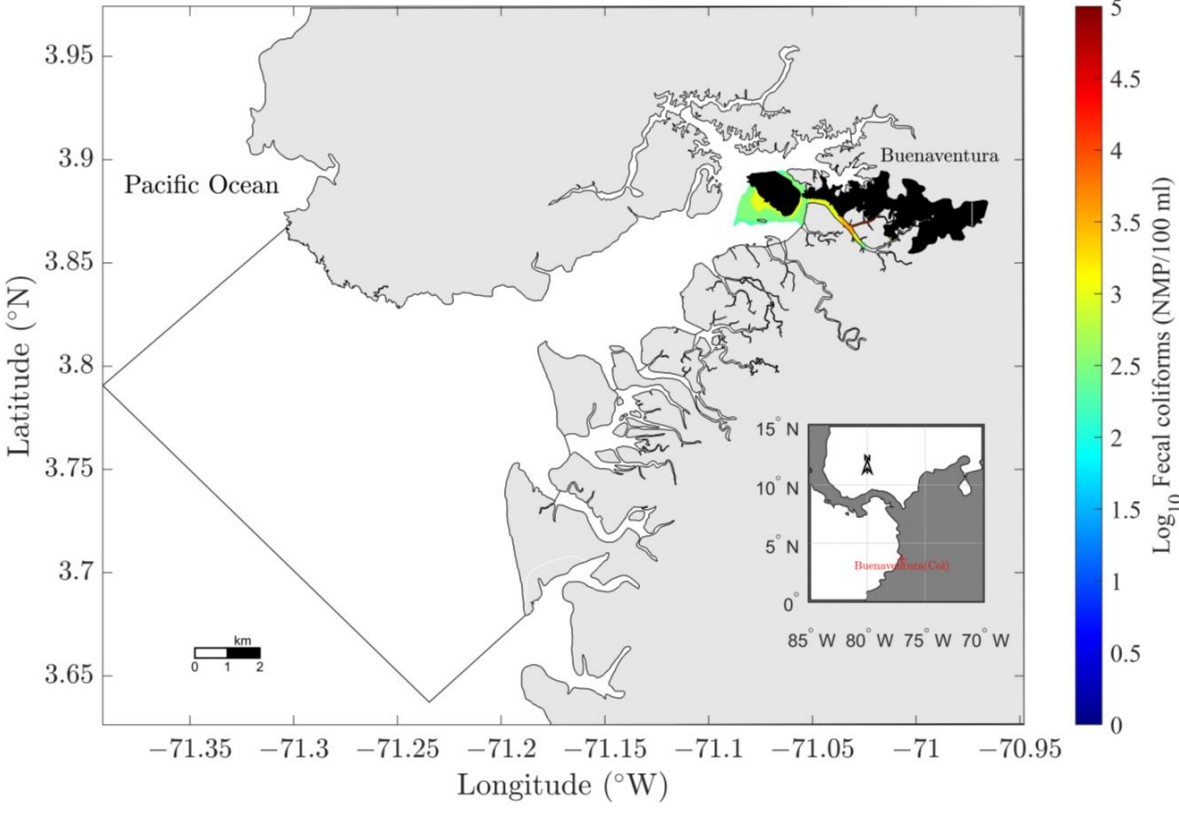

**Figure 15.** Spatial distribution of the maximum fecal coliform concentration in Buenaventura Bay in scenario four.

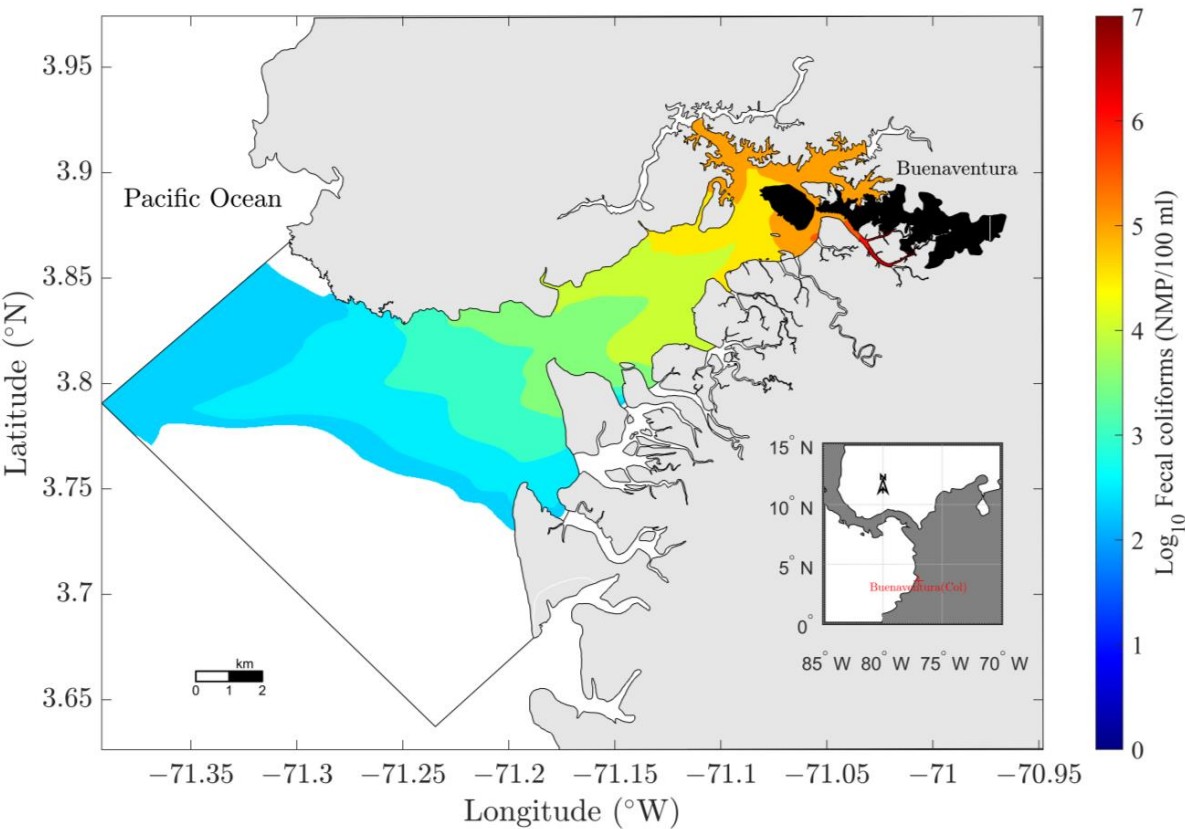

**Figure 16.** Spatial distribution of the maximum fecal coliform concentration in Buenaventura Bay in scenario five.

The study revealed that, in the absence of intervention, the concentration of FC in the bay of Buenaventura would increase significantly, with a maximum concentration of $4.32 \times 10^7$ MPN/100 mL, which is triple the current concentration. This outcome represents the highest concentration that is observed among all the scenarios analyzed and emphasizes the pressing need to implement a sanitation and discharge reduction plan. The study highlights the vital significance of taking immediate action to address the discharge of untreated wastewater in the bay of Buenaventura to avert severe environmental and health implications.

## 4. Discussion

The RMA10 3D hydrodynamic model was able to accurately simulate the tides and currents in Buenaventura Bay. The study used two statistical measures, the RMSE, and SKILL, to evaluate the model's accuracy and compared these results to field measurements. Despite the calibration and validation measurements being taken outside the bay, the model was able to successfully replicate the semi-diurnal tide and tidal range by up to 5 m in the bay during both low and high rainfall periods [4].

The study found that the prevailing current direction in Buenaventura Bay was towards the West (45%) and East (38%), with maximum currents of up to 30 cm/s, which is consistent with previous studies conducted by Otero [4]. Although a 2D model could have been used to model the hydrodynamic conditions, a 3D model was employed in this study to incorporate temperature and salinity fields at the open boundaries, which are critical variables for assessing the survival rate of FC: the variable of interest that was analyzed in the water quality model. Due to the well-mixed nature of the estuary, a 3D model was deemed appropriate even though there were no significant changes in the water column. The calibration and validation of the hydrodynamic model excluded temperature and

salinity to streamline its implementation. However, the presence of FC, the most effective indicator of wastewater in Buenaventura Bay, was verified.

No significant differences were found in the fecal coliform samples collected during the low and high rainfall seasons in Buenaventura Bay. Although Mondragon et al. [46] reported seasonal variations in the concentration of physicochemical parameters in the bay, there has been a lack of studies investigating whether this also applies to the microbiological parameters of water quality. While the primary objective of this study was not to establish significant temporal differences in coliform concentration, it was helpful for the implementation of the mode.

The RMA10 hydrodynamic model and RMA11 water quality model were calibrated and validated, indicating their suitability for simulating Buenaventura Bay's conditions. These models can simulate the spring tide and neap tide conditions, currents, and FC concentration in the bay.

The study found that there was only a 9% increase in FC concentration when comparing scenarios one and two. The results of the hydrodynamic and water quality models indicate that reducing the 695 diluted discharges to six concentrated discharges did not result in a significant change when using the maximum concentration criterion. However, to better assess the effects of reducing discharges in the different scenarios proposed in this study, the criterion for the extension of the area of exceedance was incorporated into the analysis. Figure 17 shows the areas that were affected by concentrations exceeding the FC norm for primary contact in the Bay of Buenaventura in different scenarios. Scenario one was used as a reference point for comparison (Figure 12). Areas with a frequency of exceedance of 100% indicate that all simulation results exceeded the water quality standard (200 MPN/100 mL) for fecal coliform concentrations, indicating a high risk for primary contact activities.

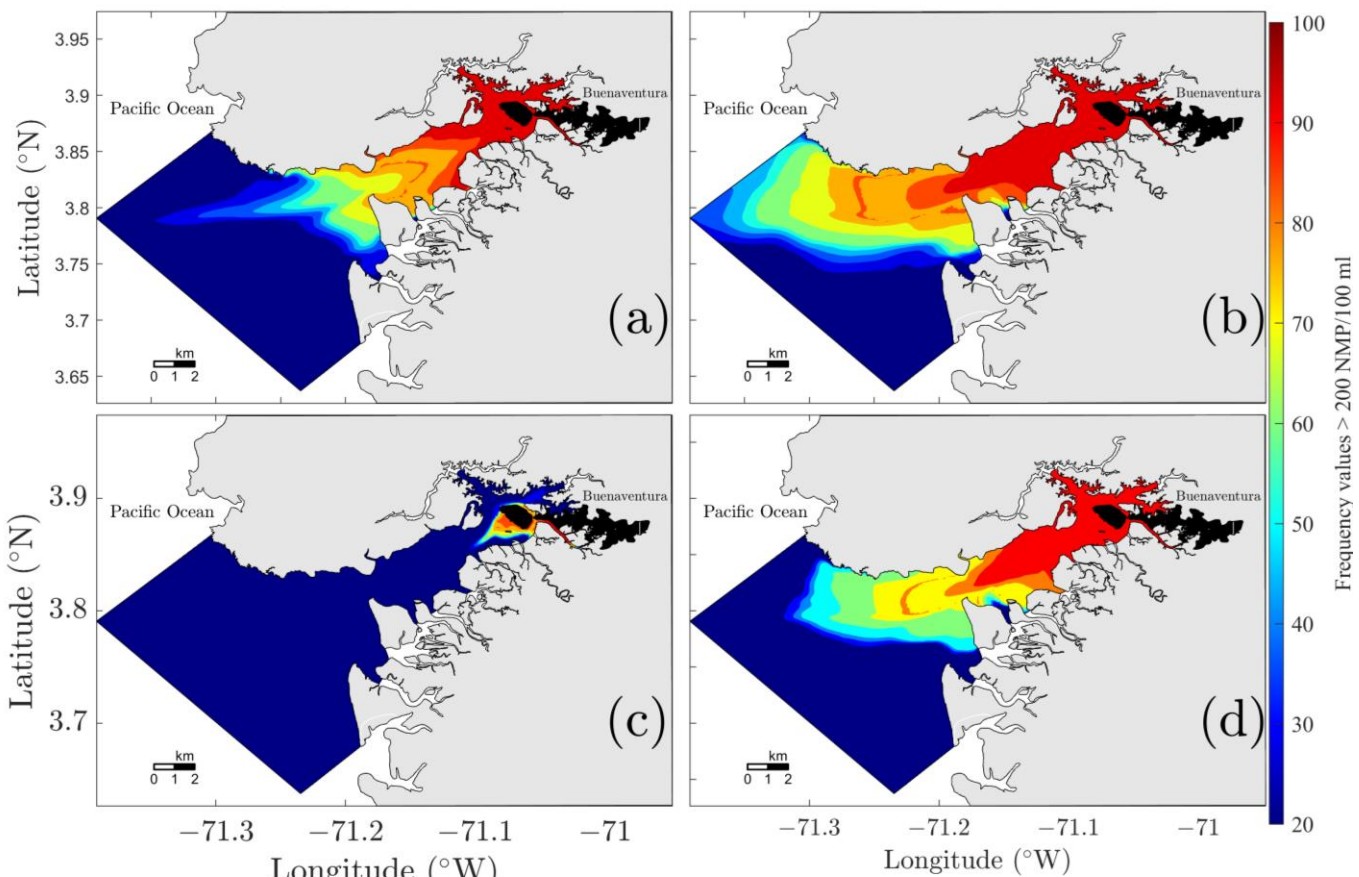

**Figure 17.** The frequency of areas affected by concentrations higher than 200 MPN/100 mL in different scenarios: (**a**) Scenario two, (**b**) Scenario three, (**c**) Scenario four, and (**d**) Scenario five.

In scenario two, which involves a reduction in discharges and the concentration of effluents at six designated points, the concentration of FC in the water exceeded the established water quality standard for primary contact, and the frequency of exceedances increased. This area expands and reaches the middle of the internal bay. Additionally, the areas that experience an exceedance frequency of between 50 and 70% also expand, extending up to the mouth of the bay (see Figure 17a).

Scenario three depicts the greatest extent of the exceedance frequency zone, in which the current discharges were concentrated at two specific points that discharged the untreated water into Buenaventura Bay (refer to Figure 17b). In this scenario, the area with a 100% frequency of exceedances extended the entire internal part of Buenaventura Bay, while the areas with a frequency of exceedance between 50% expanded beyond its external limits.

Scenarios two and three, despite having been proposed as part of the sanitation solution for Buenaventura Bay, had greater and more severe impacts than the current scenario, which involved 695 dispersed and distributed discharges throughout the bay's internal part. Scenarios two and three are necessary steps toward reaching scenario four, in which the treatment plants will be operational. However, the duration of these stages should be limited due to the amplified impact on the water quality, which will result in the reduction of 695 untreated discharges to six and, eventually, to two. Reducing the number of discharges will lead to an increase in their concentration, resulting in more significant impacts on the water quality compared to the current situation unless measures are taken to remove the contaminant load. Thus, scenarios two and three should serve as temporary solutions, and the authorities must take prompt action to meet the conditions set out in these scenarios and expedite the construction of treatment plants.

The construction of two wastewater treatment plants resulted in a significant decrease in FC concentrations in Buenaventura Bay, as demonstrated in scenario 5 (Figure 17c), where both the maximum concentrations and the frequency of exceedance decreased. However, scenario 5 (Figure 17d) illustrates how the untreated wastewater discharges still continue and where the existing discharge points continue to impact water quality in the bay.

The findings of scenario five demonstrate that without measures taken to eliminate discharges in Buenaventura Bay, FC concentrations will double, and the frequency zone exceeding the permitted limit will extend throughout the internal part of the bay.

While the construction of wastewater treatment plants is a necessary step, it alone is insufficient to fully address the contamination of the bay. Even with the treatment systems in operation, some levels of FC concentration are expected to exceed the permitted limit, as shown in the zoom of the interior of Buenaventura Bay in scenario four (Figure 18). Therefore, an additional system should be considered to resolve this issue. The PSMV proposes a submarine discharge system to disperse the concentrated discharge of the plant effluent through a diffuser system. Although this option was not studied in this analysis, the hydrodynamic and water quality models that have been developed could be used for further analysis.

This study highlights the critical water quality conditions in the Bay of Buenaventura resulting from untreated wastewater discharge. The analysis only considered the fecal coliform concentration, indicating that the situation may be even more severe than depicted. The selected sanitation strategy, if not implemented carefully and in stages, could lead to worse conditions than a scenario with no measures. Therefore, decision-makers must conduct a comprehensive analysis of the bay, considering other biological and physicochemical parameters before implementing the sanitation plan. This study provides valuable insights, but further research is necessary to supplement these findings.

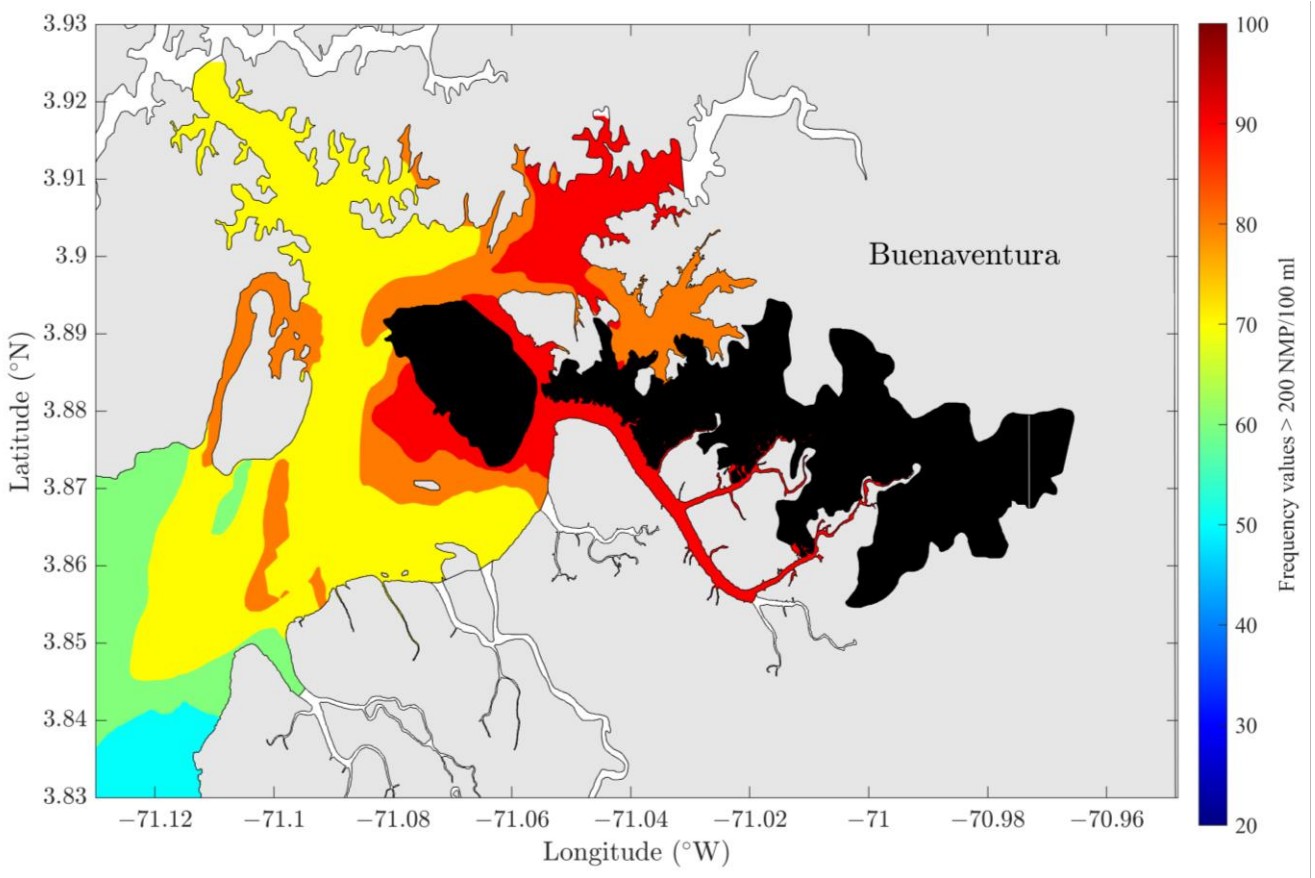

**Figure 18.** Exceeding frequency in areas affected by concentrations higher than 200 MPN/100 mL in scenario four; the inner part of Buenaventura Bay.

## 5. Conclusions

Buenaventura Bay is facing alarming pollution levels that will worsen unless sanitation actions are implemented. The Bay is an estuary that receives wastewater discharge generated by approximately 500,000 inhabitants and has resulted in the deterioration of water alongside the quality of the ecosystem and an increased risk to public health. Of particular concern are the roughly 30% of inhabitants who reside in stilt houses. This study examines the impact of a proposed 30-year sanitation plan for the estuary, which will be implemented in stages, with the ultimate goal of constructing two treatment plants. The study evaluated the effectiveness of the plan in reducing the number of discharges and improving the water quality of the ecosystem.

In order to assess the impact of domestic wastewater discharges on water bodies, the frequency of FC exceedance was used as a key indicator. This parameter, which measured the number of times that FC concentrations in the water exceeded a reference limit value, proved to be a more effective metric than simply comparing the maximum concentrations that were achieved in different scenarios. By considering both the temporal and spatial variation in concentrations that exceeded the limit value, this indicator allowed for spatial comparisons between different areas, which was affected by values that exceeded the reference standard for FC. As a result, it provided a more comprehensive and accurate evaluation of the impact of sanitation scenarios on water quality.

The intermediate stages of the sanitation plan, scenarios two and three, are more impactful than the current situation because they reduce the number of discharges but concentrate their pollutant load in six and two untreated points, respectively. It is recommended that these stages have a short duration to avoid irreversible impacts on the bay. The construction of wastewater treatment plants will bring about better sanitation conditions

by reducing FC concentrations and minimizing the frequency of exceeding the primary contact limit. Nevertheless, despite the implementation of scenario four, concerning levels of pollutant concentration still exist, and additional measures must be taken to properly dispose of effluents from the treatment plants. This information is crucial for guiding decision-making processes related to environmental management in the region.

**Author Contributions:** Conceptualization, F.-F.G.-R.; methodology, F.-F.G.-R., G.H.C. and G.A.C.N.; software, F.-F.G.-R., G.H.C. and G.A.C.N.; validation, F.-F.G.-R., G.H.C. and G.A.C.N.; formal analysis, F.-F.G.-R., G.H.C. and G.A.C.N.; writing—original draft preparation, F.-F.G.-R., G.H.C. and G.A.C.N.; writing—review and editing, F.-F.G.-R., G.H.C. and G.A.C.N. All authors have read and agreed to the published version of the manuscript.

**Funding:** This research received no external funding, and the APC was funded by F.-F.G.-R., G.H.C. and G.A.C.N.

**Conflicts of Interest:** The authors declare no conflict of interest.

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
