# Peer review of "Evaluation of Wastewater Discharge Reduction Scenarios in the Buenaventura Bay"

_water, doi:10.3390/w15061027_

Round 1
Reviewer 1 Report
1. English used all throughout the manuscript is very poor and it is very distracting.
In abstract.
2.Abstract section is very confusing and unorganised: it should begin with the statement of pollution
3.The theoretical background of calculating thermoelectric properties should be briefly introduced.
4.Authors should check the MS for grammatical errors.
5.The result is not compared with any experimental or any previous so how this result can be confirming its accuracy.
Author Response
Dear Reviewer,
Thank you for taking the time to review our manuscript and providing valuable feedback. We have carefully considered all of your comments and incorporated them into our revised manuscript.
Regarding the first comment on the poor English used throughout the manuscript, we have taken this feedback seriously and worked with a professional English editing service to ensure the language is of the highest quality. We hope that this has addressed your concerns and improved the readability of our paper.
In response to your second comment on the abstract section, we have reorganized the section to begin with a clear statement on pollution, as suggested. We believe this has helped to improve the clarity of the abstract and make it easier to understand.
We have also carefully checked the manuscript for grammatical errors, as suggested in your fourth comment.
In response to your fifth comment, we have added several references to confirm the reliability of the model used and its effectiveness in predicting scenarios for wastewater treatment and reduction of discharges into bodies of water. We believe this will help readers better understand the accuracy and reliability of our results.
Once again, we thank you for your helpful feedback and hope that these revisions have addressed your concerns. Please do not hesitate to contact us if you have any further questions or comments.
Sincerely,

Reviewer 2 Report
Expand the introduction part in the abstract.
What is the basis for the discharge types in Table 1, are the low, medium, and high discharge flowrate on average?
Please add review about RMA10 and RMA10 and SKILL for the reader’s reference.
Line 181, RM11 should be RMA11.
Should the expression in Table 1 be “Amount Discharges” or another such as “number of discharge points”?
Line 203, “corresponds to the is the mean value of observation”, please correct the sentence.
Line 233, I think you mean Table 2 not 1. Please check also Table 2 which is supposed to be 3. Check others as well.
With regards to the results from the measurements of the FC concentration that were taken at the measurement station marked as T2, are those results only from the effect of Buenaventura’s discharge to the bay? is there any discharge from the left side of the bay? you need to make your assumptions clear.
Please add the allowable limits for the FC in water.
Please include a table to show all scenarios as the current way of explanation is confusing.
Please add a flowchart for the study methodology.
Author Response
Thank you for your valuable feedback on our manuscript. We have expanded the introduction section of the abstract and made the necessary corrections to Table 1 by adding a clarification about the basis for the discharge types and using the appropriate expression. We have also included a review of RMA10 and SKILL for the reader's reference, and corrected the typo of RMA11 on line 181.
Regarding the sentence on line 203, we have revised it to correct the error. We have also checked and corrected the table numbers as you pointed out.
We appreciate your comments and have added a statement to clarify that the results of the measurements at T2 are only from the effect of Buenaventura's discharge to the bay. We have also included the allowable limits for the FC in water and added a flowchart to better explain the methodology used. Additionally, we have added a table to show all the scenarios, which we believe will make the explanation less confusing.
Thank you again for your thorough review, and we hope that our revisions meet your expectations.

Reviewer 3 Report
Dear authors,
The manuscript is an interesting study in the context of sustainable development, as it is necessary to ensure water and sanitation for the entire population according to the Sustainable Development Goals 6. Worldwide, water pollution is one of the major environmental problems and the efficiency of water pollution control is mandatory, and large investments in wastewater treatment technologies and facilities must increase in the next years.
I have the following comments:
- The manuscript requires English checking and additional proofreading. There are some incorrect formulations, the entire text should be verified (e.g.: line 296 Untreated instead of untreated; line 306 It instead of it; line 485 stud instead of study, etc.)
- What is the main objective/aim of this study, please introduce a phrase with the aim of the study and novelty. The introduction parts must be improved.
- Line 38 you should mention drinking water instead of water;
- For big numbers of four or more digits, you must use commas (lines 63- 61,162m3;) when you have a small number you must use point (e.g.: Lines 67, 69, etc). Please revise all the numbers existing in the manuscript.
- Please correct the abbreviations CF with FC (fecal coliforms), as it has been mentioned in the article (lines 453, 455, 468, and 474).
- The correct unit for BOD is mgO2/L, also for the dissolved oxygen (line 53).
- Reconsider the table number. In line 233, you have Table 2 instead of Table 1, and then you have Table 2 twice.
- The figure's number should be in the order of appearance in the text (not Figure 6, then 9, and then 7).
- You have mentioned the mean values 2.8*103MPN/100 ml and 1.8-103MPN/100ml but in the table, you have different numbers (3 respectively 1.4)
Also, you have some quotation marks incorrectly used (lines 135 and 472).
The conclusion should be improved by introducing more pertinent conclusions of the work.
Thus, my decision is a MAJOR REVISION.
Author Response
Point 1: The manuscript requires English checking and additional proofreading. There are some incorrect formulations, the entire text should be verified (e.g.: line 296 Untreated instead of untreated; line 306 It instead of it; line 485 stud instead of study, etc.)
Response 1: We have made significant improvements to the language and writing of our manuscript. As a result, we have submitted it for English language editing services offered by MPDI.
Point 2: What is the main objective/aim of this study, please introduce a phrase with the aim of the study and novelty. The introduction parts must be improved.
Response 2: The study objective was descripted, See Line 79-85
Point 3: Line 38 you should mention drinking water instead of water;
Response 3: Ok, the correction has been made. Please see lines 38-39.
Point 4: For big numbers of four or more digits, you must use commas (lines 63- 61,162m3;) when you have a small number you must use point (e.g.: Lines 67, 69, etc). Please revise all the numbers existing in the manuscript.
Response 4: All number existing in the manuscript were revised and corrected
Point 5: Please correct the abbreviations CF with FC (fecal coliforms), as it has been mentioned in the article (lines 453, 455, 468, and 474).
Response 5: The abbreviations of Fecal coliform was corretec
Point 6: The correct unit for BOD is mgO2/L, also for the dissolved oxygen (line 53).
Response 6: The units of BOD and DO were corrected, see line 53-54
Point 7: Reconsider the table number. In line 233, you have Table 2 instead of Table 1, and then you have Table 2 twice.
Response 7: In the manuscript, Tables 2 and 3 were both numbered and referenced accurately
Point 8: The figure's number should be in the order of appearance in the text (not Figure 6, then 9, and then 7).
Response 8: The figures were appropriately numbered in the manuscript.
Point 9: You have mentioned the mean values 2.8*103MPN/100 ml and 1.8-103MPN/100ml but in the table, you have different numbers (3 respectively 1.4)
Response 9: the values in table 3 were corrected
Point 10: Also, you have some quotation marks incorrectly used (lines 135 and 472).
Response 10: The manuscript was corrected
Point 11: The conclusion should be improved by introducing more pertinent conclusions of the work.
Response 11: : The conclusion was improved

Round 2
Reviewer 2 Report
The authors have responded adequately to the comments on their previous submission. However, the following comments need some attention:
- The authors need to change the titles of "Table 3. Concentration of FC in wastewater discharge" and "Table 4. Concentration of FC in the wastewater discharge" as they should have different descriptions.
- Please make this statement more clear in Line 379-380, "In the central part of the bay, high FC levels were observed between 50% and 70% of the time". What is the 50-70% of the time?
Author Response
Dear Reviewer,
I am writing to inform you that we have addressed the comments you made in your latest review of our article. Specifically, we have made the following changes:
We have added a note to clarify that the data in the table refer to the maximum, minimum, and mean values of FC in the water column of Buenaventura Bay for the FC measurements (see lines 301-302). Finally.
We have also clarified that the time referenced in the model results is based on 35,040 results with a time step of 15 minutes for a one-year run in 2021. (see line 380-381).
Thank you for your valuable feedback, which has helped us to improve our article.
Sincerely,
Francisco García
Reviewer 3 Report
Dear author,
The article has been improved but needs moderate English changes in the text. When you answer different questions to the reviewer you have to mention the lines from the new version of the article. Otherwise, it is very difficult to follow the changes.
Best regards.